

# Exploring the potential of the RPA system SUMO for multi-purpose boundary layer missions during the BLLAST campaign

Joachim Reuder[1], Line Båserud[1], Marius O. Jonassen[1,2], Stephan Kral[1], and Martin Müller[3]

[1]Geopysical Institute, University of Bergen, and Bjerknes Centre for Climarte Research, Allegaten 70, 5007 Bergen, Norway
[2]UNIS - The University Centre in Svalbard, 9171 Longyearbyen, Norway
[3]Lindenberg und Müller GmbH & Co. KG, Hohenhameln, Germany

*Correspondence to:* Joachim Reuder (joachim.reuder@uib.no)

**Abstract.** In June and July 2011 the RPA system SUMO performed a total number of 299 scientific flights during the BLLAST campaign in Southern France. Three different types of missions have been performed, vertical profiling of the mean meteorological parameters temperature, humidity and wind, horizontal surveys of the surface temperature and horizontal transects for the estimation

of turbulence. The manuscript provides an introduction to the corresponding SUMO operations, including regulatory issues, and the coordination of manned and unmanned airborne operations for boundary layer research that have been pioneered during the BLLAST campaign.

The main purpose of the SUMO flight strategy was atmospheric profiling in high temporal resolution. A total of 168 profile flights have been performed during the campaign with typically more than

10 flights per Intensive Observational Period (IOP) day. The collected data allow for a detailed study of boundary layer structure and dynamics and will in the future also be used for further analysis, e.g. the determination of profiles of sensible and latent heat fluxes. First tests of a corresponding method have shown very promising results and have provided surface flux values in close agreement with those from ground based eddy-covariance measurements. In addition 74 horizontal surveys of the

IR emission of the surface have been performed at altitudes of around 65 m. Each of those surveys covers typically an area of around 1 km$^2$ and allows for an estimation of the surface temperature variability, an important information for the assessment of the heterogeneity of the surface forcing as function of soil and vegetation properties. The comparison with other surface temperature measurements shows that the raw data of the airborne and ground observations can differ considerably,

but that even a very simple multiple regression method can reduce those differences to a large degree. Finally also 49 flight missions for the measurement of velocity variance have been realized during



the BLLAST campaign. For that SUMO has been equipped with a 5 hole probe (5HP) sensor for the determination of the flow vector with 100 Hz. In particular for this application there is still need for further improvement, both with respect to the aircraft and sensor hard- and software, and the al-
gorithms and methods for data analysis and interpretation. Nevertheless have the SUMO operations during the BLLAST campaign shown the vast potential of small and light-weight RPA systems with low infrastructural demand for atmospheric boundary layer research.

## 1   Introduction

The atmospheric boundary layer (ABL) and in particular the surface layer (SL) directly adjacent to
the ground are characterized by large spatial and temporal variability, especially over non-homogeneous land-use and terrain. The required resolution in space and time for an appropriate characterization and investigation of a wide range of related phenomena by classical boundary layer instrumentation, e.g. meteorological masts and towers or ground based remote sensing profilers as sodars, radars or lidars, is therefore logistically demanding and not easy to achieve.
Beginning with the pioneering work of Konrad et al. (1970) in the 1970s boundary layer researchers have started to use remotely controlled aircraft for atmospheric measurements (e.g., Egger et al., 2002, 2005; Spengler et al., 2009; Reuder et al., 2011). With the availability of reasonably priced and sized autopilot systems a considerable number of of fixed and rotary-wing airframes of different size, endurance and complexity, have found their way into atmospheric research during the
last decade (e.g., Holland et al., 2001; Shuqing et al., 2004; Spiess et al., 2007; Elston et al., 2011; Wildmann et al., 2014). A comprehensive summary and overview has recently been compiled and published by Elston et al. (2015).
    The atmospheric boundary layer (ABL), with a typical vertical extension in the order of hundreds of m to a few km is a natural target for measurements with relatively small and light-weight
RPA (Remotely Piloted Aircraft) systems of limited payload capacity and endurance. The rapid development in the field of micro-electronics and micro-electromechanical systems (MEMS) during the last years has provided smaller, faster and more energy-efficient sensors, both for atmospheric measurements and the attitude determination and control of the airframes. As a consequence of this miniaturization even very small and lightweight RPA systems, with a take-off weight below 1kg, are
now capable to carry various sensors for multiple BL measurements.
    One of those systems is the Small Unmanned Meteorological Observer (SUMO), a collaborative development between the Geophysical Institute at the University of Bergen (GFI/UiB), Norway and Lindenberg und Müller GmbH & Co. KG, a small enterprise specialized in unmanned system development for atmospheric research in Hildesheim, Germany. A detailed description of the basic
SUMO system and its ongoing development can be found in Reuder et al. (2009, 2012). The mechanical properties of the aircraft are open source, and the blueprints and building instructions are



freely available. Besides GFI/UiB, that owns and operates at the time being 4 SUMO systems, have also several other institutes worldwide started to use it during the last years. To the knowledge of the authors these are the Finnish Meteorological Institute (FMI), the University Centre in Svalbard (UNIS), the Universities of Oklahoma (OU) and Boulder (UC) in the United States, and ETH in Zürich, Switzerland.

In the past the SUMO system has mostly been used for ABL profiling missions to investigate various local scale meteorological phenomena, often in combination with meso-scale numerical simulations. Some examples are studies on terrain induced flow modification at the Hofsjökull glacier on Central Iceland (Mayer et al., 2010), on the potential benefit of assimilating SUMO data into the numerical weather forecast model WRF to improve the short term prediction capabilities (Jonassen et al., 2012), a detailed study of the polar BL in Adventdalen on Svalbard (Mayer et al., 2012) or the investigation of the effect of Nunataks on local meteorology in Antarctica (Stenmark et al., 2014). A large portion of the SUMO operations has been taking place in polar regions (e.g., Cassano, 2014; Mayer et al., 2012; Jonassen et al., 2015), documenting the robustness, flexibility and low infrastructural demands of the SUMO system. In this context the hand-starting capability of the SUMO system has proven its huge benefit compared to the larger airframes that require starting aid in form of a catapult or at least a bungee cord, the latter being negatively affected both by very low and very high temperatures. A modified SUMO airplane has been equipped with a specifically designed 7 hole flow probe that has been developed by ETH in Zürich, Switzerland (Kocer et al., 2011; Subramanian et al., 2015) for the purpose of turbulence measurements in the wake of wind turbines. The integration of a 5 hole probe (5HP) on the SUMO systems of GFI (Reuder et al., 2012; Båserud et al., 2014) is also partly motivated by that purpose, but will of course also be beneficial for ABL research in general.

The main objective of the study is the presentation of the capability of the SUMO system to perform a wide range of specific and targeted flight missions and to give an indication of the quality and application of the data sets obtained. By that we can describe the added value of such missions in the framework of a large atmospheric boundary layer campaign.

The manuscript is organized as follows. Section 2 gives a short overview of the SUMO airframe and sensor payload used during the BLLAST (Boundary-Layer Late Afternoon and Sunset Turbulence) campaign. A general description of the campaign with focus on the RPAS operations, including regulatory issues and the coordination between manned and unmanned aircraft operations, is presented in Section 3 together with the detailed description of the different types of flight missions of the SUMO system. Exemplary results are introduced and discussed in Section 4 before the manuscript ends with a short conclusion and outlook.





## 2 The SUMO system

The Small Unmanned Meteorological Observer (SUMO) is a micro-RPAS with a length and a wingspan of 80 cm and a take-off weight of around 650 g (Reuder et al., 2009) and has been continuously improved and developed during the last years (Reuder et al., 2012). The main differences

in airframe, autopilot and meteorological sensor package (Table 1) between the version described in Reuder et al. (2012) and the system finally deployed during the BLLAST campaign are described in the following.

### 2.1 Airframe

The SUMO airframe is based on the commercially available model aircraft kit FunJet by Multiplex,

which has been reinforced by glass fiber coating at the bottom of the fuselage, the front part of the wing and the base for the motor at the rear part of the fuselage. These modifications enhance the aircrafts stiffness and resistance against damage from landings on rough surfaces, thus improving the airframe's durability and also flight performance. The top of the entire fuselage can be removed to allow easy access to the sensors and electronics inside.

### 2.2 Autopilot and control units

The autopilot system in use is Paparazzi, an open source Hardware and software autopilot system developed and maintained under the lead of the École Nationale de l'Aviation Civile, Toulouse, France (Brisset et al., 2006; ENAC, 2008). A new compact and lightweight IMU (Inertia Measurement Unit) system has been integrated for the measurement of the aircraft's attitude as an replacement for

the previously used IR sensor array. This extends the operation of SUMO to a much wider range of atmospheric conditions, such as flight missions under low or even within clouds, which could not be performed before, since the IR based attitude control required a certain minimum temperature difference between sky and ground to work properly. Besides the extended applicability this also improved the reliability and the overall performance of the autopilot and provides us with more ac-

curate measurements of the pitch and roll angles, which are required for the calculation of turbulent parameters from the turbulence measurement system.

### 2.3 Basic meteorological parameters

The basic meteorological parameters pressure, temperature and humidity are measured by an integrated set of sensors as summarized in Table 1. The temperature and humidity sensor are mounted

on top of the wings inside radiation protection tubes and are well ventilated during flight missions. The pressure sensor is mounted inside the fuselage for better protection and is combined with an additional temperature sensor, which is used for monitoring the thermal state of the battery and electronic components, information of particular importance for operations in hot or cold environments.

| Parameter | Sensor | Range | Accuracy | Acquisition frequency |
|---|---|---|---|---|
| Temperature | Sensirion SHT75 | −40 to 124 °C | ±0.3 °C | 2 Hz |
| Humidity | Sensirion SHT75 | 0 to 100 % | ±2 % | 2 Hz |
| Temperature | PT1000 Heraeus M222 | −32 to 96 °C | ±0.2 °C | 8.5 Hz |
| Pressure | MS 5611 | 300 to 1200 hPa | | 4 Hz |
| Surface temperature | MLX90614 | | | 8.5 Hz |
| 3D flow vector | 5 hole probe (5HP), Aeroprobe | 11 to 35 ms$^{-1}$ | ±0.1 ms$^{-1}$ | 100 Hz |

**Table 1.** Specifications of the meteorological sensors.

A second temperature sensor (PT1000) with a response time of about 1 s has been implemented for
an improved temperature–height assignment during profile measurements.

### 2.4 Turbulence

The integration of a 5HP and the corresponding pressure transducers and data logger (Aeroprobe,
2012; Reuder et al., 2012), enables the measurement of the 3D flow vector in the Lagrangian system
of the RPAS at a temporal resolution of 100 Hz, a sufficient resolution for the calculation of turbu-
lence parameters, such as the turbulence kinetic energy ($TKE$) or the turbulent momentum flux ($\tau$),
also referred to as Reynolds stress. The probe is mounted in the nose of the airframe and is connected
to the differential pressure sensors in the logging unit by six silicon tubes of about 10 cm length. The
tip of the sensor is located approximately 10 cm in front of the fuselage. The key specifications of
the turbulence sensor are also listed in Table 1. During the BLLAST campaign the Aeroprobe system
has not yet been fully integrated into the SUMO's data acquisition system so that two different data
loggers were in use, inducing additional challenges for the post processing of the turbulence data due
to unsynchronized data loggers. This is described in more detail in Båserud et al. (2015). Another
problem related to determining turbulence parameters is related to the lack of a precise method to
measure the aircraft's yaw angle, which has to be known for the transformation of the measured flow
field into an earth-fixed coordinate system. In multi-copters typically magnetometers are applied for
this purpose, however they do not provide a very precise angle estimation. In addition the small
dimension of the SUMO system does not allow for a sufficient separation distance between the mag-
netometer and the cables connecting battery and motor. Therefore the magnetic field, induced by the
electric current required for powering the motor during flight, will be a further source of uncertainty.

### 2.5 Surface temperature

A downward looking infrared (IR) sensor, mounted in one of the wings, can be used to give an
estimate of the surface temperature. The angle of view of this sensor is 90°, so that the measured





radiation originates from a circular footprint area with a diameter that is equal to the flight level above ground, assuming horizontal flight with zero pitch and roll. Therefore, the calculated surface temperatures have to be regarded as a mean temperature estimate of the footprint area. The information on the long-wave radiation, emitted by the surface, can be converted to a corresponding surface temperature by applying the Stefan-Boltzmann law and corresponding corrections for atmospheric absorption and emission in the atmospheric layer between the sensor and the surface.

## 3    The BLLAST campaign

The BLLAST field campaign (Lothon et al., 2014) was conducted from 14 June to 8 July 2011 at and around Lannemezan in southern France. A wide range of ground-based and airborne instrumented platforms have been deployed and used, including remote-sensing profilers, radiosondes, tethered balloons, surface flux stations and meteorological towers, as well as manned full-size aircraft and RPAS. By that the boundary layer structure and evolution, from the earth's surface to the free troposphere, was heavily monitored during the entire day, with a particular focus on the time period between noon and sunset.

The main purpose of the BLLAST campaign is the investigation of the turbulence decay during the afternoon transition from a fully developed and highly turbulent convective boundary layer (CBL) towards evening conditions characterized by the transformation of the CBL into the less turbulent residual layer and the growing of a stable boundary layer (SBL) from the ground. A particular focus of the campaign was directed to the effects of heterogeneity on this transition, including both the local scale, e.g. small scale variability of surface and vegetation properties, and the meso-scale, e.g. the topography of the Lannemezan site in the proximity of the Pyrenees, leading to persistent thermally-driven flow patterns in the area under certain synoptic conditions.

### 3.1    RPAS operations

The BLLAST campaign was, to the knowledge of the authors, the first ABL campaign to use coordinated operations between manned and small remotely piloted aircraft systems as an integrated measurement strategy for the probing of the ABL and relevant properties of the underlying surface. The main intention was to close the observational gap between the fixed and local scale measurements by ground based in-situ and remote sensing instrumentation and the fast moving sensors used for observations of the regional scale on the two participating manned aircraft, a Piper Aztec from SAFIRE in France (Saïd et al., 2005), and a Sky Arrow from CNR in Italy (Gioli et al., 2006). It turned out that RPA systems, in particular SUMO with its small size and low weight, and a resulting hand-start capability, provided a flexible and fast solution for atmospheric profiling that among others was highly beneficial for the mission planning of the manned aircraft operations.



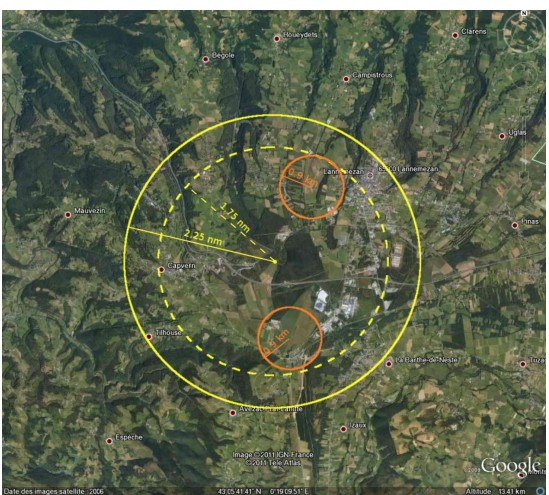

**Figure 1.** The temporary restricted area activated for the RPAS operations during the BLLAST campaign. The solid yellow line indicates the outer boundary of the 2.25 nautical mile (nm) zone reserved, the dashed one the 1.75 nm finally available for the RPAS operations due to an additional required safety buffer towards general airspace. The two small circles indicate the zones around the Sites 1 and 2 where all RPAS operations took place.

Basis for the successful operation of different RPAS during the BLLAST campaign was a collaborative effort between the French Civil Aviation Authority (DGAC), the operators of the Lannemezan observatory and the individual groups flying RPAS, upfront and during the campaign. In a first step a Temporary Restricted Airspace (TRA) has been established. The area is a cylinder with a radius of

185     2.25 nautical miles (nm) centered at 43° 6'18" N and 0° 21'6" E (see Fig. 1) that covers both heavily instrumented Sites 1 and 2 of the campaign (small orange circles in the figure). The solid yellow line indicates the outer boundary of the 2.25 nm zone, the dashed one the 1.75 nm finally available for the RPAS operations due to an additional required horizontal safety buffer towards general airspace. The overall vertical extension of the TRA was 7500 ft. Considering a mandatory 500 ft safety buffer

190     on top and the surface elevation of ca. 600 m in the area, the vertical range of RPAS operations was constraint to an altitude of appr. 1500 m above ground. During the campaign the TRA was generally activated for 16 hours per day from 05 UTC to 21 UTC. In a second step each RPAS group had to apply individually to DGAC for an individual flight permission. In general both coordinated multiple RPAS missions and beyond line of sight (BLOS) operations had been approved during the

195     campaign, e.g. allowing SUMO to probe frequently the whole available vertical range of the TRA. During multiple RPAS missions at Site 1, an appointed RPAS coordinator was responsible for the coordination of the different groups and the internal segregation of the RPAS airspace to avoid the danger of collisions.





During parallel operations with the manned research aircraft, the RPAS coordinator was also in direct radio contact with the pilots. Particular limitations for RPAS operations during the activation hours of the TRA applied for time periods with an active flight plan for the two manned research aircraft, that had a special permission to enter the TRA. Both the Sky Arrow and the Piper Aztec have a typical endurance of ca. 2 h for a scientific mission. Depending on the strategy of each individual IOP with manned aircraft participation, of using one or both aircraft and for the latter case either flying sequentially or in parallel, RPAS missions have been altitude limited for a period between 2 to 8 hours during the afternoon of the corresponding day. In those situations RPAS and manned aircraft had to keep a vertical separation of at least 500 ft. In practice this meant that all RPAS had to stay at any time of the manned aircraft flight at least 500 ft below the lowest approved flight level for that mission.

Besides SUMO, two other RPA systems participated during the whole BLLAST campaign, the Meteorological Mini Aerial Vehicle ($M^2AV$) Carolo from the University of Braunschweig (Martin et al., 2011), and the Multipurpose Airborne Sensor Carrier (MASC) from the University of Tübingen (Wildmann et al., 2014) (see Fig. 2). Both systems are particularly suited for flying kilometre-scale level legs performing high-frequency measurements of wind components, temperature and humidity fluctuations (e.g., van den Kroonenberg et al., 2012; Wildmann et al., 2013) and therefore for the determination of heat, momentum and moisture fluxes. The MASC system, participating in a prototype version, suffered from technical problems and no data sets could be supplied to BLLAST. A number of other RPAS only participated during the last two weeks of the BLLAST campaign (for details see Lothon et al. (2014)). These adjunct operations were performed as a RPAS test and sensor intercomparison event, organized by the European COST Action ES0802 Unmanned Aerial Systems in Atmospheric Research (Lange and Reuder, 2013). The main scientific contributions of those RPAS operations were Octocopter flights performed by the University of Applied Sciences Ostwestfalen-Lippe for micro-scale air and surface temperature surveys (Cuxart et al., 2015) and orthofoto flights with the RPAS Sirius (University of Heidelberg and Karlsruhe Institute of Technology) providing an areal camera survey of the Sites 1 and 2 with a resolution of better than 5 cm.

### 3.2 SUMO operations during the campaign

Three identical SUMO airframes were operated during the BLLAST campaign. They successfully completed a total of 299 individual flight missions on 23 days in the period between 13.06. and 08.07.2011 (see Table 2). The majority of the flights was performed during the 12 Intensive Operational Periods (IOPs) of the experiment (Lothon et al., 2014). Three different types of scientific missions have been conducted, namely atmospheric profiles (168 flights), areal surveys (74 flights), and turbulence transects (49 flights) and are described in more detail in the following. The remaining 8 flights were dedicated to system tests and autopilot tuning.



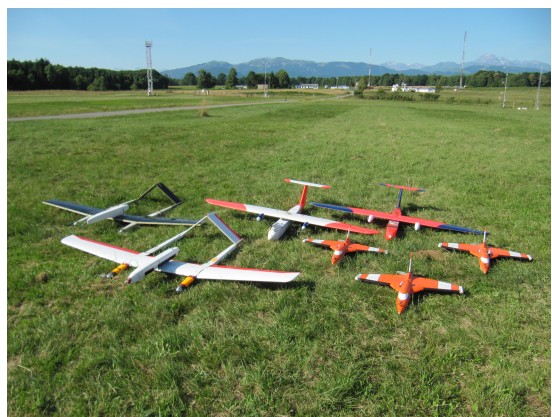

**Figure 2.** The fleet of RPAS operated during the whole BLLAST campaign at Site 1 in Lannemezan. From left to right: MASC (University of Tübingen), $M^2AV$ (University of Braunschweig) and SUMO (University of Bergen). In the background the Pyrenees and the 60 m instrumented tower in the upper right corner.

### 3.2.1 Profiles

235 The profile missions are performed as helical flight patterns along the wall of an imaginary cylinder with a diameter of 120 m (see Fig. 3). The maximum flight altitude was limited by the vertical extension of the TRA to 7000 ft, corresponding to approximately 1500 m above the ground level of slightly above 600 m. Due to energy efficiency considerations, SUMO was operated with a rather high ascent speed of ca. 7-10 $\mathrm{ms}^{-1}$. During the descent, usually in gliding mode without engine, 240 the vertical velocity is considerably lower in the order of 2-3 $\mathrm{ms}^{-1}$. The data acquisition rate for temperature and humdity of 2 Hz results in a vertical resolution of the corresponding profiles in the order of 5 m for the ascent and 1 m for the descent. The flight pattern provides 2 consecutive profiles of the atmospheric parameters within a time window of about 15 minutes. The time constants of the temperature and humidity sensors in use, typically in the order of several seconds, result in slightly 245 shifted profiles during ascent and descent. Assuming stationary conditions, this can be numerically corrected (Jonassen, 2008), in addition also providing information on the mainly temperature dependent magnitude of the sensor time constants.

### 3.2.2 Areal surveys

The areal surveys consist of consecutive parallel transects in East-West and North-South direction in 250 a distance of 150 m (see Fig. 4). With the capacity of one battery pack (2600 mAh), approximately 1 $\mathrm{km}^2$ could be covered by one flight. The flights should be performed at rather low altitudes to minimize longwave absorption and emission from the atmospheric layer between the infrared (IR) sensor on SUMO, used to monitor the surface temperature, and the ground. For the BLLAST campaign al-



**Table 2.** SUMO operations during the BLLAST campaign.

| Date | IOP | Total | Test | Profile | Survey | Turbulence |
|---|---|---|---|---|---|---|
| 13.06. | - | 7 | 3 | | | 4 |
| 14.06. | 0 | 3 | | | 1 | 2 |
| 15.06. | 1 | 22 | 1 | | 2 | 19 |
| 16.06. | - | 1 | 1 | | | |
| 17.06. | - | 11 | | 7 | 2 | 2 |
| 18.06. | - | 5 | | 5 | | |
| 19.06. | 2 | 28 | | 12 | 13 | 3 |
| 20.06. | 3 | 23 | | 11 | 10 | 2 |
| 21.06. | - | 8 | | 8 | | |
| 23.06. | - | 2 | | 2 | | |
| 24.06. | 4 | 12 | | 10 | 2 | |
| 25.06. | 5 | 23 | 1 | 11 | 9 | 2 |
| 26.06. | 6 | 25 | 2 | 11 | 8 | 4 |
| 27.06. | 7 | 35 | | 12 | 12 | 11 |
| 30.06. | 8 | 18 | | 12 | 6 | |
| 01.07. | 9 | 11 | | 6 | 5 | |
| 02.07. | 10 | 17 | | 14 | 3 | |
| 03.07. | - | 6 | | 6 | | |
| 04.07. | - | 9 | | 9 | | |
| 05.07. | 11 | 14 | | 13 | 1 | |
| 06.07. | - | 7 | | 7 | | |
| 07.07. | - | 8 | | 8 | | |
| 08.07. | - | 4 | | 4 | | |
| Total | | 299 | 8 | 168 | 74 | 49 |

titudes between 65 and 80 m were chosen to ensure sufficient vertical clearance from buildings, trees
and the 60-m meteorological mast at Site 1. At an altitude of 60 m, the angle of view of the infrared
sensor of 90 deg results in a circular sensor footprint at the ground of 60 m diameter (assuming zero
pitch and zero roll), over which the temperature information is averaged. In the post-processing of
the IR data, all data points in which the aircraft had a pitch and/or roll angle larger than 10 degrees
have been filtered out from the dataset. This was done to avoid temperature signals far afield from
the area of interest. All the IR data from Site 1 have been corrected for an emissivity of 0.95 and
for Site 2, the emissivity was set to 0.97, corresponding to the different surface conditions at the two
sites.




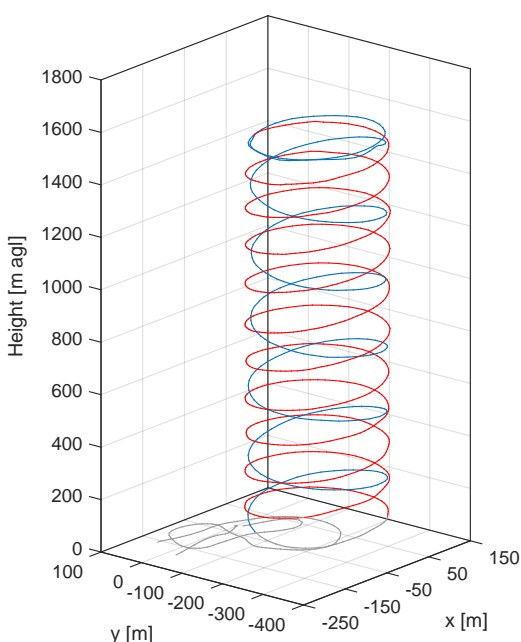

**Figure 3.** Typical atmospheric profiling flight pattern during the BLLAST campaign. The ascent is colored in blue, the descent in red.

### 3.2.3 Turbulence transects

The flight patterns for the turbulence transects consist of straight legs of ca. 1 km length with turning circles on both ends (see Fig. 5). Only the straight leg, colored in red, is used for the determination of the turbulence parameters. Depending on the ambient wind speed and direction, the typical flight time for a single 1 km leg was in the order of 35 to 60 s. Usually 4 straight legs, 2 in each direction, have been performed at one altitude before climbing to the next level. Following this pattern, the battery capacity allowed for a maximum of 4 altitudes to be covered during one flight. These altitudes have been chosen as 65, 130, 300 and 500 m above ground. Some flights covered only a subset of those altitudes and some flights were also covering a higher level of 1000 m.

### 4 Results

This section will give an overview of the observational data set obtained by the different types of SUMO flight missions during the BLLAST campaign. Given the large number of flights this has to be done by implication on the basis of examples. Nevertheless are we confident that the presentation and discussion of those examples will provide a good picture of the information content of the





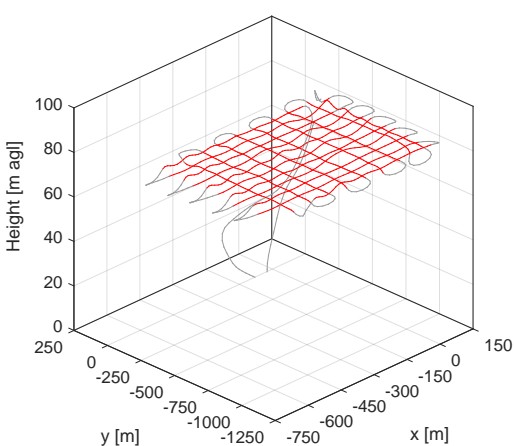

**Figure 4.** Typical flight pattern for the surface temperature surveys during the BLLAST campaign. Data are only evaluated for the straight flight legs colored in red, where the infrared sensor is assumed to look vertically down.

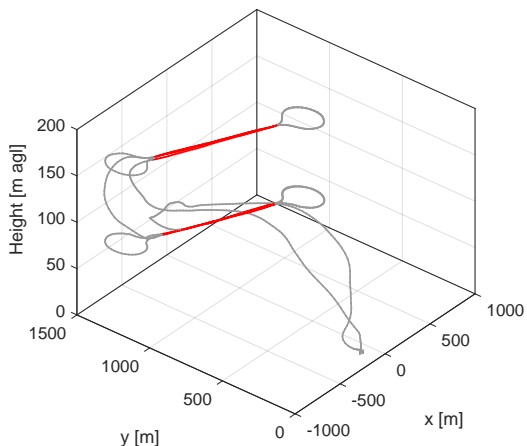

**Figure 5.** Typical flight pattern for the turbulence measurements during the BLLAST campaign. Turbulence data are only evaluated for the straight flight legs colored in red.

SUMO dataset that is already under investigation (e.g. Båserud et al., 2015; Cuxart et al., 2015; Lothon et al., 2014; Pietersen et al., 2015) and will be studied in even more detail in the future.



### 4.1 Atmospheric profiling

The vertical structure of the ABL was the target of more than half of the flight missions during the BLLAST campaign. During each IOP SUMO typically provided more than 10 profiles over the day (see Table 2 and contributed therefore considerably to the monitoring of the diurnal development of the CBL in high temporal resolution. The short pre-flight preparation time of the SUMO system in the order of 10 min allowed for a fast and flexible measurement program providing in-situ

observations of the vertical structure of the atmosphere in near real time. The corresponding information from flights performed prior to the daily morning briefing was important decision support for the further measurement program of an IOP, e.g. for detailed planning of the flight levels of the manned aircraft missions or for the vertical distribution of sensors and the measurement strategy of the tethered balloon deployments.

Figures 6 and 7 present two examples of the diurnal development of the ABL for June 19 (IOP 2) and June 21 (a non-IOP day), two days with rather different BL structure and dynamics. The first flight at 06 UTC on June 19 shows a shallow stable layer at the ground and a well-mixed residual layer above up to an altitude of 1200 m. During the day the whole BL is warming and moistening and the capping inversion is continuously descending, reaching a level of around 700 m at 21 UTC.

This is a clear signature of synoptic-scale dynamic subsidence. The last profile shows again the development of a surface inversion and a slight stabilizing tendency in the residual layer above. The free atmosphere (FA) is over the day warming by ca. 5 K and the specific humidity $q$ increases from values of below 1 $\mathrm{gkg}^{-1}$ to more than 7 $\mathrm{gkg}^{-1}$, both indicating also a considerable influence of warm air advection. The profiles taken around noon and in the early afternoon exhibit also a

higher local variability both in temperature and humidity within the capping inversion, suggesting the presence of small-scale processes modifying the entrainment zone. June 21 is characterized by an in general relatively shallow ABL and Fig. 7 presents the development for a 5 h period around noon. The first ascent of SUMO at 09 UTC shows a CBL of around 300 m depth capped by an extended stable layer up to 800 m. Until 14 UTC the CBL warms by 4 K and finally grows to around

600 m where it merges with the top of the very stable layer that simultaneously subsides. The FA above is slightly cooling, indicating weak cold air advection. The moisture increases from around 10 to 12 $\mathrm{gkg}^{-1}$ in the BL and from 7 to 8 $\mathrm{gkg}^{-1}$ in the FA.

The temporal resolution of the SUMO flights, typically in the order of one hour, allows for a detailed analysis of the BL structure based on profiles fixed over one position, in contrast to e.g.

the radiosoundings performed that measure slant profiles along the balloons trajectory. One example for such an advanced analysis is the estimation of turbulent flux profiles of sensible and latent heat from the profiles of the corresponding mean quantities $\theta$ and $q$. The algorithm in use here has been developed and applied for observations by the RPAS SMARTSonde and is described in detail in Bonin et al. (2012), based on a method suggested by Deardorff et al. (1980). It is in general based

on a simplified version of the prognostic equation for $\theta$ or $q$ and allows to relate the change of the





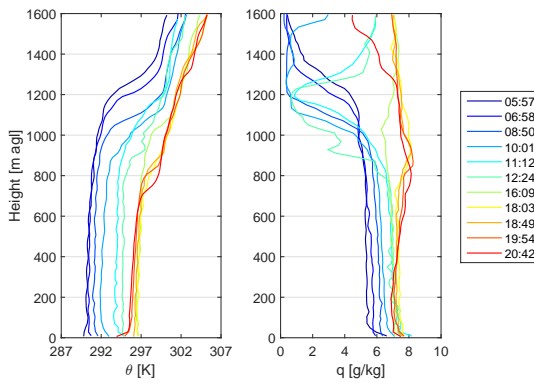

**Figure 6.** Profiles of potential temperature $\theta$ and specific humidity $q$ measured by SUMO on 19.06.2011.

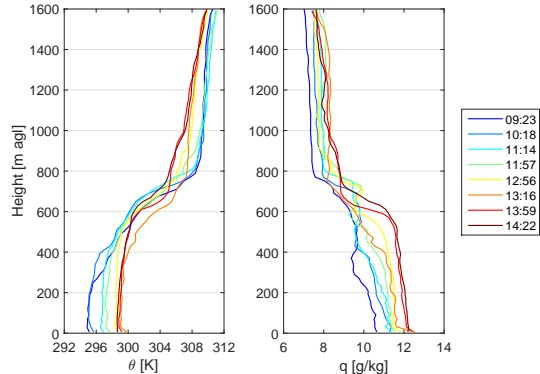

**Figure 7.** Profiles of potential temperature $\theta$ and specific humidity $q$ measured by SUMO on 21.06.2011.

mean quantity with time to the corresponding flux divergence. Figure 8 presents two examples for calculated sensible heat flux profiles for July 5. The profiles of $\theta$ (left panel) show a growth of the CBL from 400 m at 09:17 UTC to around 500 m at 10:29 UTC and finally ca. 800 m at 12:09 UTC. The sensible heat flux profiles (right panel) follow the expected shape in the CBL with a linear

decrease with height and slightly negative values on top of the BL due to entrainment processes. The retrieved ground values of around 120 $\mathrm{Wm}^{-2}$ fit very well with the observations from the network of the eddy covariance stations for that time (Lothon et al., 2014). A thorough analysis of all available SUMO profiles under the aspect of flux profile is in progress and expected to result soon in a publication of its own.




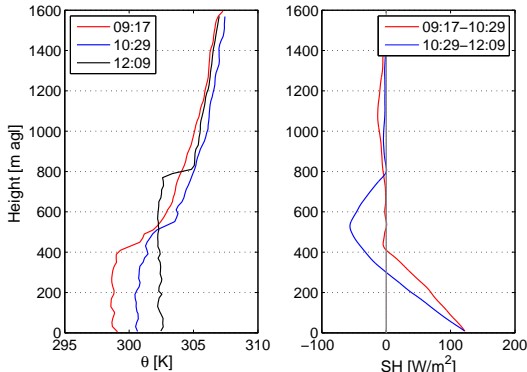

**Figure 8.** Three subsequent profiles of potential temperature $\theta$ taken on July 5 (left panel) and the corresponding calculated profiles of sensible heat flux (right panel).

### 4.2 Areal surveys

The areal surveys of the surface temperature were intended to provide supplementary information for the investigation of the heterogeneous surface forcing in selected regions of the campaign area, both for the CBL during daytime and SBL during the night. Such data are expected to improve the interpretation of the point measurements by the surface energy balance stations, e.g. with respect to spatial representativeness. The BLLAST campaign offered the possibility to test the potential of an airborne low-cost IR sensor for surface temperature estimates against other well established measurement methods like net radiometers. In using the net radiometers, the upwelling long wave radiation has been converted into surface temperature using the following formula $T_s = \sqrt[4]{\frac{LW_{out}}{\epsilon\sigma}}$ where $T_s$ is the surface temperature, $LW_{out}$ is the upwelling long wave radiation, $\epsilon$ is the emissivity and $\sigma$ is the Stefan-Boltzmann constant ($5.67 \times 10^{-8} W m^{-2} K^{-4}$). For Site 2, an $\epsilon$ of 0.97 has been chosen, corresponding to the surface types there.

In the period June 14 - July 5, 24 areal surveys were performed at Site 1. Figures 9 and 10 show examples of surface temperatures measured by SUMO at Site 1 during daytime and in the evening. Amongst the differences that stand out between the two cases is the fact that the small heterogeneity site (marked by a red box) is relatively warm during daytime and relatively cold during the night. The soil in this rectangular area has in the past been compacted as foundation for a radar antenna array. As a consequence of the resulting structural changes, such as increased density and lower soil moisture content, this area warms and cools distinctly faster than its surroundings. The observed differences in the order of a few K can be expected to have an impact on local circulations and thus microclimate (e.g. Cuxart et al., 2015).

In the period June 25 - 27, a total of 28 areal surveys were performed at Site 2. Figure 11 shows an example of the surface temperature field derived from SUMO measurements at Site 2 during



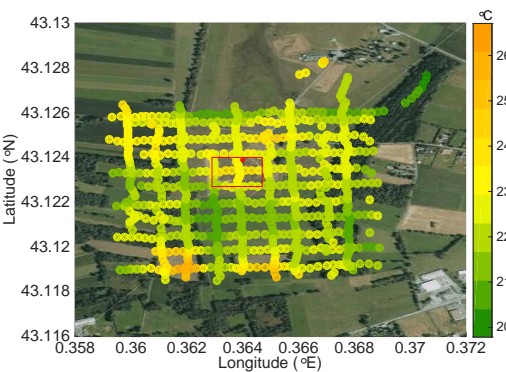

**Figure 9.** Surface temperature from Site 1 on 19.06.2011 at 19:41 UTC measured using the downward looking SUMO IR sensor. Data are only shown for the straight flight legs, where the IR sensor is assumed to look vertically down. The red square marks the location of the 'small scale heterogeneity site'.

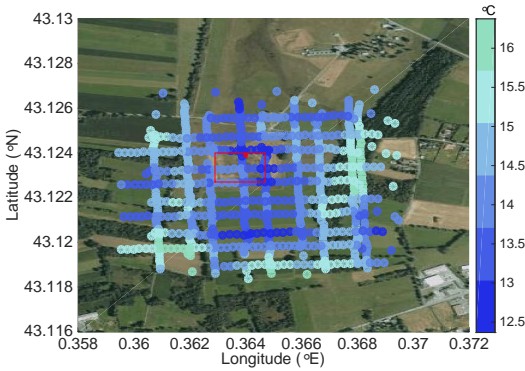

**Figure 10.** Same as in Fig. 9, but for 02.07.2011 at 17:43 UTC.

daytime. Four areas, expected to represent main characteristic surface types, are marked by colored boxes. Moor - characterized by a mixture of bare soil and sparse vegetation (blue), corn - a cultivated
corn field with a canopy height of around 120 cm during the measurement (red), forest - a Douglas fir canopy with a typical tree height of 20-25 m (green), and waste site - an industrial area of waste disposal with a mixture of dry gravel, asphalt and buildings (yellow). The surface temperatures within the boxes for the four different characteristic areas have been averaged for each survey flight and the corresponding time series is presented in Fig. 12.
Amongst the four surface types, the forest clearly has the coldest infrared temperature signature in the beginning of the day and until about noon. This can be recognized in both the horizontal survey data (Fig. 11) and in the time series (Fig. 12). This changes towards the night and in the latest



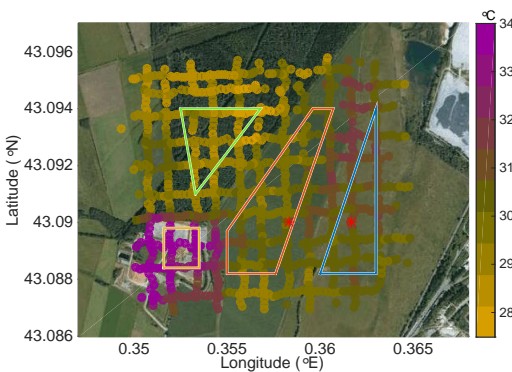

**Figure 11.** Surface temperature from Site 2 on 27.06.2011 at 13:15 UTC measured using the downward looking SUMO IR sensor. Data are only shown for the straight flight legs, where the IR sensor is assumed to look vertically down. Four areas with characteristic surface types are marked with coloured boxes: moor (blue), corn (red), forest (green), waste site (yellow). The locations of two net radiometers (AWS) within the moor and corn areas are marked as well.

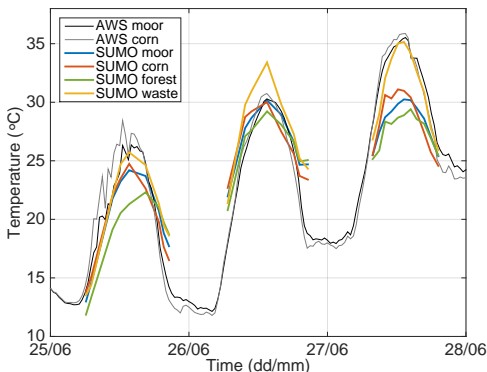

**Figure 12.** Time series of surface temperature from four different surface types at Site 2 measured using the downward looking SUMO infrared sensor and two net radiometers.

measurements at about 20:30 UTC only the waste site is about as warm or warmer than the forest. The waste site has the highest maximum temperature in the SUMO data set, reaching almost 35°C

on June 27. The differences in surface temperature between the moor and corn fields are subtle, but clear. During night-time corn is the cooler, while in the morning and until about noon it is the warmer amongst the two. This goes along with the fact that corn heats up the quickest in daytime and it also cools down the quickest in the night-time. These findings are confirmed when considering the AWS





surface temperature observations from the corn and moor fields, which were obtained from Kipp &
Zonen CNR1 net radiometers.

When comparing the SUMO and AWS data, it is clear that they at times deviate considerably in
both directions. At times SUMO being clearly colder (e.g. mid-day of June 25 and 27) or showing
roughly the same temperature (June 26). At some occasions, the SUMO measurements are even
slightly warmer, e.g. in the nights of June 25 and 26. Several factors may contribute to such differ-
ences when comparing the ground based radiometers and the airborne IR sensor on SUMO. One
factor that has to be taken into account is the fact that the AWS measurements only represent point
measurements that are limited in space, and that presumably are strongly dominated by the local
conditions. The RPAS data, on the other hand, are averaged over considerably larger areas and de-
rived from footprints that already are averages over ca. 10000 $\text{m}^2$. Another, probably even more
important factor, is the distance between sensor and surface. The AWS measurements were made
at of 2 m (moor) and 2.8 m (corn) above the ground, while the SUMO measurements were per-
formed at an average elevation of around 75 m (Site 1) and 69 m (Site 2). It is well known, e.g. from
the retrieval of sea surface temperatures using satellites, that parts of the infrared radiance from the
surface is attenuated and potentially re-emitted by the atmosphere before it reaches the radiometer
(e.g. Kilpatrick et al., 2001). Corrections for atmospheric effects are therefore done for IR satellite
data on a routine basis. Amongst the atmospheric constituents, water vapor has been found to have
the strongest effect in this regard. Different correction algorithms have been proposed to correct for
atmospheric effects (e.g. Grassl and Koepke, 1981; Holyer, 1984). In this study, we have in a first
step used a simple correction based on a multiple linear regression algorithm using the following
predictor variables: the IR surface temperature measured by the AWS, specific humidity and air
temperature measured by SUMO and the SUMO altitude. The regression was performed on 6 inde-
pendent flights from June 26 and 27 in which straight legs (transects) were flown at two different
elevations (around 60 and 150 m) above the moor. The results of this simple correction method are
presented in Fig. 13.

It can be seen that the agreement between the AWS and SUMO based surface temperature esti-
mates is now clearly improved. Taking the AWS data as reference, the root mean square error in the
SUMO data is reduced from 3.0°C to 1.9°C when comparing the original data with those obtained
by the linear regression algorithm. Although there is uncertainty remaining, this shows the general
potential of the SUMO IR measurements for surface temperature retrieval. Future studies are re-
quired to improve and fine-tune the simple algorithm described above or to apply a more advanced
correction algorithm. The BLLAST dataset provides several further opportunities of corresponding
improvement and validation, e.g. by consideration of further data sets, as e.g. from a net-radiometer
(Eppley-PIR, model 29435F3) operated by the Physical-meteorological Observatory Davos World
Radiation Centre (PMOD/WRC) (Gröbner et al., 2009) on the small scale heterogeneity field at Site
1, or from the thermal IR camera (FLIRA320) operated on the 60 m tower by the University of





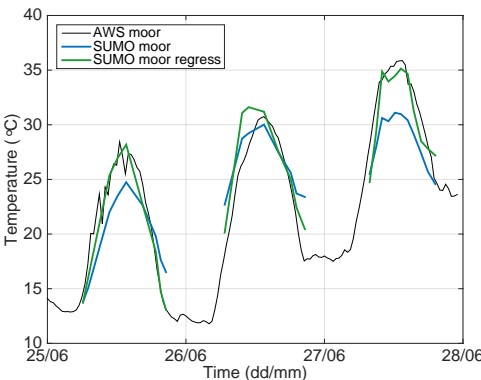

**Figure 13.** Timeseries of surface temperature from the moor area using a net radiometer, unaltered SUMO data and corrected SUMO data using the multiple linear regression algorithm.

California, San Diego (Garai et al., 2013). This is however more than enough material for a separate publication and clearly outside the scope of this paper.

### 4.3 Turbulence

The BLLAST campaign was one of the first real world applications after adapting the 5HP based
turbulence system to SUMO and has therefore mainly to be seen as test and validation campaign for that purpose. Technical imperfections, as the use of two different data loggers for the 5HP data and the aircraft attitude required for the motion compensation, as well as different sampling rates for both data sets, required an elaborate pre-processing of the data. Further details can be found in Båserud et al. (2015). Another issue that occurred during the campaign was an instability in the
altitude control of the autopilot system not handled correctly by the motion compensation algorithm, leading to artificial modulations in the vertical velocity component (see lower panel of Fig. 14) in the 100 Hz time series. The horizontal components $u$ and $v$ are completely unaffected by this feature.

During June 19 and 20 four turbulence transect flight missions have been performed in the vicinity of the 60 m tower that was equipped with a sonic anemometer (Campbell CSAT-3) on the top level
for comparison purposes. The corresponding SUMO flight altitude was between 65 and 75 m. The energy spectra for the velocity components $u$, $v$, and $w$ for flight # 29 on June 19 around 16 UTC are presented in Fig. 15. They show for the horizontal components a good agreement in the energy level between the SUMO and the sonic data sets, except for a region of enhanced energy in the SUMO spectra at around 1 Hz. This is most likely related to a frequency of the internal attitude control of the
autopilot system that creates additional motions that are misinterpreted as real atmospheric motions. This feature has to and will be further investigated in the future. For the vertical component SUMO follows again over a wide range of frequencies the expected -5/3 slope of the inertial subrange, but





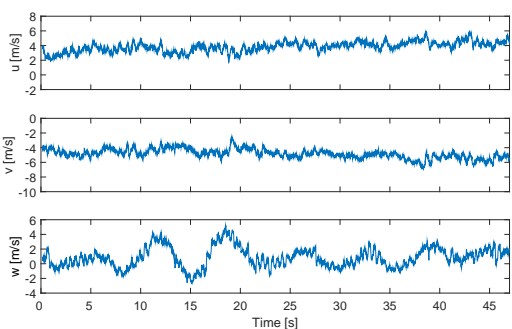

**Figure 14.** Example for the time series of the wind components $u$,$v$, and $w$ in the meteorological coordinate system derived from the SUMO 5HP system in a temporal resolution of 100 Hz. Presented are the data for one single leg of flight # 29 on 19.06.2011 between 15:50:37 and 15:51:23 UTC.

shows an in general too high overall energy level. The peak around 0.1 Hz is clearly related to the modulations in the vertical velocity shown in the lower panel of Fig. 14) and described before. The

reason for the overall shift towards higher energy content is not yet clear and requires also further investigation.

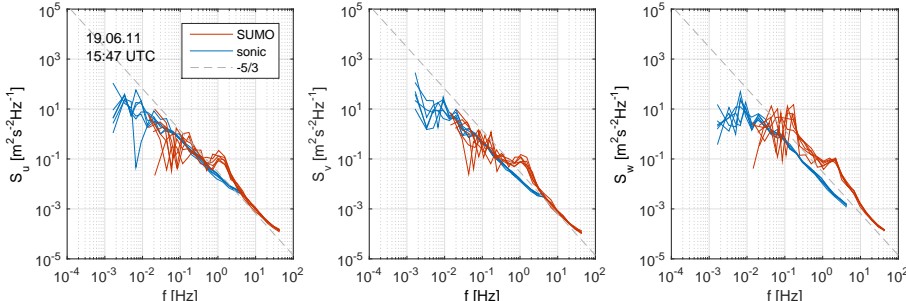

**Figure 15.** Energy spectra of the velocity variances of the $u$,$v$, and $w$ component from SUMO (red) and the sonic anemometer at the 60 m mast. The data are from flight # 29 on 19.06.2011 around 16 UTC. The average wind speed was $3.6\ \mathrm{ms}^{-1}$ from $317°$.

    Figure 16 presents the energy spectra for flight # 31 on June 20 around 16 UTC. It shows in general the same behaviour for the SUMO system, but reveals an interesting feature for the sonic anemometer at the 60 m tower. With a wind direction from Northeast the sonic anemometer is

obviously located partly in the mast shadow, leading to an enhanced energy content in all three velocity components for this instrument.



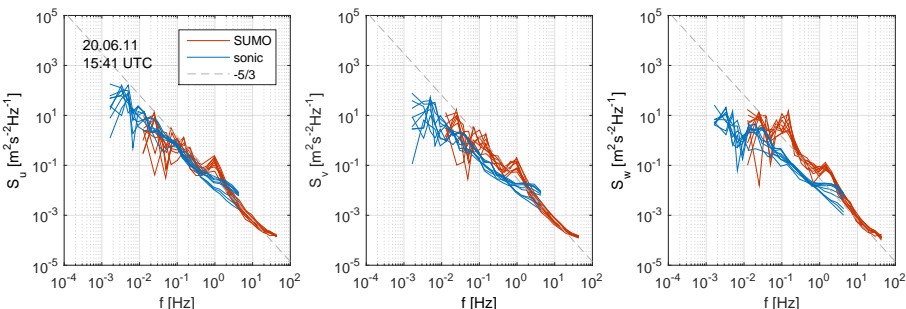

**Figure 16.** Energy spectra of the velocity variances of the $u$, $v$, and $w$ component from SUMO (red) and the sonic anemometer at the 60 m mast. The data are from flight # 31 on 20.06.2011 around 16 UTC. The average wind speed was 2.7 ms$^{-1}$ from 53°.

## 5 Summary and outlook

Operations of RPAS were a substantial part of the ABL measurement strategy during the BLLAST campaign in June and July 2011. Several fixed wing and rotary wing systems contributed with mea-
surements of atmospheric parameters and relevant surface properties. With multiple RPAS operations and the coordinated operations of manned and unmanned research aircraft BLLAST was without doubt a milestone in the application of such systems for atmospheric research.

The experience with the application procedure for the required flight permissions was in general very positive. Good communication, mutual understanding between the aviation authorities and the
RPAS research groups and a bit of pragmatism were the key for success in this context. The available airspace for RPAS operations in the TRA up to about 1500 m above ground was in most cases sufficient to probe the relevant parts of the ABL and the lower FA to achieve the scientific goals of the campaign.

The RPA system SUMO performed a total number of 299 scientific flights during the BLLAST
campaign. Three different types of missions have been performed, vertical profiling of the mean meteorological parameters temperature, humidity and wind, horizontal surveys of the surface temperature, and horizontal transects for the estimation of turbulence parameters. The main purpose of the SUMO flight strategy was atmospheric profiling in high temporal resolution. A total of 168 profile flights have been performed during the campaign with typically more than 10 flights per IOP
day. The collected data allow for a detailed study of boundary layer structure and dynamics and will in the future also be used for further analysis, e.g. the determination of profiles of sensible and latent heat fluxes. First tests of a corresponding method have shown very promising results and have provided surface flux values in close agreement with those from ground based eddy-covariance measurements. A follow on study and publication, dedicated specifically to this scientific question, is in
preparation. In addition 74 horizontal surveys of the IR emission of the surface have been performed





at altitudes of around 65 m. Each of those surveys covers typically an area of around 1 km$^2$ and allows for an estimation of the surface temperature variability, an important information for the assessment of the heterogeneity of the surface forcing as function of soil and vegetation properties. The comparison with other surface temperature measurements shows that the raw data of the airborne
and ground observations can differ considerably, but that even a very simple multiple regression method can reduce those differences to a large degree. A more detailed analysis of the acquired surface temperature data, including improved correction algorithms for atmospheric absorption and re-emission, and a comparison and validation by consideration of further data sets, as e.g. from a net-radiometer (Eppley-PIR, model 29435F3) and from a thermal IR camera (FLIRA320), is also
planned. 49 flight missions for the measurement of velocity variance have been realized during the BLLAST campaign. For that SUMO has been equipped with a 5HP sensor for the determination of the flow vector with 100 Hz. In particular for this application there is still need for further improvement, both with respect to the aircraft and sensor hard- and software, and the algorithms and methods for data analysis and interpretation. The main technical challenges to be addressed in this context
are the logging of the aircraft attitude from the IMU sensors with 100 Hz and a reliable method for the determination of the yaw angle of the aircraft during flight that is required for a accurate motion correction.

*Acknowledgements.* The BLLAST field experiment was made possible thanks to the contribution of several institutions and supports: INSU-CNRS (Institut National des Sciences de l'Univers, Centre national de la
Recherche Scientifique, LEFE-IDAO program), Météo-France, Observatoire Midi-Pyrénées (University of Toulouse), EUFAR (EUropean Facility for Airborne Research) and COST ES0802 (European Cooperation in Science and Technology). The field experiment would not have occurred without the contribution of all participating European and American research groups, which all have contributed to a significant amount.
The BLLAST field experiment was hosted by the instrumented site of Centre de Recherches Atmosphériques,
Lannemezan, France (Observatoire Midi-Pyrénées, Laboratoire dAérologie). BLLAST data are managed by SEDOO, from Observatoire Midi-Pyrénées.
The participation of the Meteorology Group of the Geophysical Institute, University of Bergen was facilitated by contributions of the Geophysical Institute and the Faculty of Mathematics and Natural Sciences under the "små driftsmidler" scheme, a travel stipend by the Meltzer Foundation in Bergen, and the Short Term Scien-
tific Mission (STSM) scheme within the COST Action ES0802 "Unmanned Aerial Vehicles in Atmospheric Research".
The authors are grateful to Anak Bhandari for the technical assistance in the preparation of the campaign, and to Christian Lindenberg, the SUMO chief pilot during BLLAST. Without his passion, determination and patience we would never have achieved this large number of flights.



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
