# Peer review of "Exploring the potential of the RPA system SUMO for multi-purpose boundary layer missions during the BLLAST campaign"

_Atmospheric Measurement Techniques, 2015_

## Referee Comment (RC1) · J. Cassano (Referee) · 1 Feb 2016

This manuscript describes boundary layer observations made by a small remotely piloted aircraft system (RPAS) during the BLLAST campaign. The manuscript provides a brief summary of previous RPAS use for boundary layer studies, describes the SUMO RPAS and instrumentation in detail, discusses the SUMO observational strategy and issues related to securing airspace during BLLAST, and provides examples of observations acquired with the three main flight patterns (profiles, aerial surveys, and turbulence transects). The manuscript is well written and the content is an excellent fit for AMT. I recommended acceptance of this manuscript with only minor revisions.

Specific comments
[Figure]

**AMTD**

Line 60: Should be University of Colorado not University of Boulder.

Line 332: Should this sentence refer to longwave radiometer instead of net radiometer since the remainder of the sentence refers to LWout?

---

## Referee Comment (RC2) · Anonymous Referee #2 · 7 Feb 2016

General Comments: This manuscript provides an overview of current efforts to make atmospheric measurements using the UAS SUMO. This is an excellent fit for AMT, as it provides insight into novel technology for performing atmospheric research. In general, the manuscript is well-written, clear, and informative. Figures are readable and clear, and it is obvious that the authors have spent a good amount of time to prepare this paper. I have very few criticisms of the work, and therefore suggest that the paper be published with minor revisions.

Specific comments:

- Line 60: This should say "University of Colorado – Boulder (CU)". Note that the correct abbreviation is CU, not UC (don't ask me why...)
[Figure]

- Line 147: Can you provide some additional information on the IR sensor? What is the technology? How big is it?

- Line 184: Recommend replacing "has been" with "was"

- At some points, there may be a bit too much detail. For example, while the paragraph form lines 199-209 is very interesting, I think that the authors should consider whether all of these details are necessary here (I recommend reading through the manuscript one more time to assess whether the level of detail is appropriate, keeping in mind that the readership is likely looking for information on sensors and measurements, and perhaps are less interested in the fine details of operations).

- Lines 245-246: Has there been any consideration for performing corrections similar to those applied for radiosondes (e.g. Miloshevich et al., 2004)

- Line 266: how were the turbulence legs oriented to the prevailing wind?

- Line 408: Pre-processing? Or post-processing?

---

## Editor Comment (EC1) · E. Pardyjak (Editor) · 7 Apr 2016

Dear Dr. Reuder, The responses to the reviewers look very reasonable. Please also define RPA and SUMO in the abstract.

Also, throughout the document place commas after prepositional phrases.

Thanks! Eric

---

## Author Comment (AC1) · 7 Apr 2016

First of all we would like to thank both reviewers for their review and very positive and constructive feedback. In the following the reviewers comments have been listed together with our response.

Reviewer 1

Line 60: Should be University of Colorado not University of Boulder. changed

Line 332: Should this sentence refer to longwave radiometer instead of net radiometer since the remainder of the sentence refers to LWout?

This has now been formulated clearer: "The BLLAST campaign offered the possibility to test the potential of an airborne low-cost IR sensor for surface temperature estimates against other well established measurement methods like pyrgeometers or 4-component net radiometers. In using the output of the radiometers, the upwelling longwave radiation has been converted. . .."

Reviewer 2

Line 60: This should say "University of Colorado – Boulder (CU)". Note that the correct abbreviation is CU, not UC (don't ask me why...)

Changed, see also reviewer 1

Line 147: Can you provide some additional information on the IR sensor? What is the technology? How big is it?

Corresponding information has been added and the paragraph in the manuscript reads now as: A downward looking infrared (IR) sensor, MLX90614 produced by Melexis, mounted in one of the wings, can be used to give an estimate of the surface temperature. The sensor consists of a thermopile detector chip sensitive for infrared radiation and a signal processing unit integrated in a TO-39 housing, i.e. a small metal cylinder with 8.2 mm diameter and 4.1 mm length.

Line 184: Recommend replacing "has been" with "was".

changed

At some points, there may be a bit too much detail. For example, while the paragraph form lines 199-209 is very interesting, I think that the authors should consider whether all of these details are necessary here (I recommend reading through the manuscript one more time to assess whether the level of detail is appropriate, keeping in mind that the readership is likely looking for information on sensors and measurements, and perhaps are less interested in the fine details of operations).

We have considered the comment on the high level of detail in the paragraph of the flight operations and the coordination between RPAS and manned aircraft operations. As the BLLAST campaign was the first coordinated operation of both measuring platforms for atmospheric research, we feel that it is appropriate and important to keep the paragraph as it is.

Lines 245-246: Has there been any consideration for performing corrections similar to those applied for radiosondes (e.g. Miloshevich et al., 2004).

We are aware of the corrections proposed by Miloshevich and others for radiosonde data. The Miloshevich method requires extensive laboratory characterizations of the sensors to perform the corrections. Having both data from ascent and descent within a very short time interval (typically 10 minutes) and at the exact same position from our SUMO flights, we are confident that we can perform an appropriate correction by the methods used and described and without relying on the laboratory data.

Line 266: how were the turbulence legs oriented to the prevailing wind? This was highly variable throughout the 49 turbulence flights performed and cannot be answered in general here.

Line 408: Pre-processing? Or post-processing?

We decided to call it just "processing", as both expressions would fit, "pre-processing" with respect to that we have to do it before calculating the turbulence parameters, "post-processing" in the context as it is done not on-board the aircraft, but after the flight.

In addition to the reviewers comments have two references in the manuscript, Båserud et al., 2016 and Cuxart et al., 2016 been updated corresponding to the actual status of the manuscripts.

Please also note the supplement to this comment:

http://www.atmos-meas-tech-discuss.net/amt-2015-397/amt-2015-397-AC1-supplement.pdf

[Figure]

**Supplement:**

Manuscript prepared for Atmos. Meas. Tech.
with version 2014/07/29 7.12 Copernicus papers of the LaTeX class copernicus.cls.
Date: 7 April 2016

[revised manuscript text omitted]
 sensor consists of a thermopile detector chip sensitive for infrared radiation and a signal processing unit integrated in a TO-39 housing, i.e. a small metal cylinder with 8.2 mm diameter and 4.1 mm length. The angle of view of this sensor is $90°$, so that the measured radiation originates from a circular footprint area with a diameter that is equal to the flight level above ground, assuming horizontal flight with zero pitch and roll. Therefore, the calculated surface temperatures have to be regarded as a mean temperature estimate of the footprint area. The information on the long-wave radiation, emitted by the surface, can be converted to a corresponding surface temperature by applying the Stefan-Boltzmann law and corresponding corrections for atmospheric absorption and emission in the atmospheric layer between the sensor and the surface.

**3 The BLLAST campaign**

The BLLAST field campaign (Lothon et al., 2014) was conducted from 14 June to 8 July 2011 at and around Lannemezan in southern France. A wide range of ground-based and airborne instrumented platforms have been deployed and used, including remote-sensing profilers, radiosondes, tethered balloons, surface flux stations and meteorological towers, as well as manned full-size aircraft and RPAS. By that the boundary layer structure and evolution, from the earth's surface to the free troposphere, was heavily monitored during the entire day, with a particular focus on the time period between noon and sunset.

The main purpose of the BLLAST campaign is the investigation of the turbulence decay during the afternoon transition from a fully developed and highly turbulent convective boundary layer (CBL) towards evening conditions characterized by the transformation of the CBL into the less turbulent residual layer and the growing of a stable boundary layer (SBL) from the ground. A particular focus of the campaign was directed to the effects of heterogeneity on this transition,

[revised manuscript text omitted]

---

## Author Comment (AC2) · 18 May 2016

SUMO and RPAS has been defined in the abstract and the commas behind prepositional phrases have been included throughout the text
* * *